# First Report of the Emerging Pathogen *Kodamaea ohmeri* in Honduras

**DOI:** 10.3390/jof10030186

**Published:** 2024-02-28

**Authors:** Bryan Ortiz, Roque López, Carlos Muñoz, Kateryn Aguilar, Fernando Pérez, Isis Laínez-Arteaga, Fernando Chávez, Celeste Galindo, Luis Rivera, Manuel G. Ballesteros-Monrreal, Pablo Méndez-Pfeiffer, Dora Valencia, Gustavo Fontecha

**Affiliations:** 1Instituto de Investigaciones en Microbiología, Universidad Nacional Autónoma de Honduras, Tegucigalpa 11101, Honduras; bryan.ortiz@unah.edu.hn (B.O.); kateryn.aguilar@unah.edu.hn (K.A.); fernandoperez@unah.hn (F.P.); jchavezv@unah.hn (F.C.); lrivera@unah.edu.hn (L.R.); 2Laboratorio Nacional de Vigilancia, Secretaría de Salud de Honduras, Tegucigalpa 11101, Honduras; rohlome@gmail.com (R.L.); carlosmicro1728@hotmail.com (C.M.); 3Laboratorio de Bacteriología, Hospital Mario Catarino Rivas, San Pedro Sula 21101, Honduras; izla04@gmail.com; 4Departamento de Microbiología, Instituto Hondureño de Seguridad Social, Tegucigalpa 11101, Honduras; celestegalindom@gmail.com; 5Departamento de Ciencias Químico, Biológicas y Agropecuarias, Universidad de Sonora, Campus Caborca, Caborca 83600, Mexico; manuel.ballesteros@unison.mx (M.G.B.-M.); pablo.mendez@unison.mx (P.M.-P.); dora.valencia@unison.mx (D.V.)

**Keywords:** *Kodamaea ohmeri*, Honduras, virulence, diagnosis, adhesion, macrophages

## Abstract

*Kodamaea ohmeri* is an environmental yeast considered a rare emerging pathogen. In clinical settings, the correct identification of this yeast is relevant because some isolates are associated with resistance to antifungals. There is a lack of available data regarding the geographical distribution, virulence, and drug resistance profile of *K. ohmeri*. To contribute to the knowledge of this yeast, this study aimed to describe in depth three isolates of *K. ohmeri* associated with fungemia in Honduras. The identification of the isolates was carried out by sequencing the ribosomal ITS region. In addition, the susceptibility profile to antifungals was determined, and some properties associated with virulence were evaluated (exoenzyme production, biofilm formation, cell adhesion, and invasion). The isolates showed strong protease, phospholipase, and hemolysin activity, in addition to being biofilm producers. Adherence and invasion capacity were evident in the HeLa and Raw 264.7 cell lines, respectively. This study expands the understanding of the underlying biological traits associated with virulence in *K. ohmeri*, and it is the first report of the detection and identification of *K. ohmeri* in Honduras as a cause of human infection.

## 1. Introduction

Fungi represent one of the most varied kingdoms of eukaryotes, and it is believed that they constitute between 1.5 and 5 million fungal species [1], with nearly 300 species of fungi associated with infections in humans [2,3]. The selection, adaptation, and dissemination of novel fungal infections have been influenced by multiple factors in recent years, including climate change, the widespread use of antifungals in agriculture, and large-scale migrations [4,5,6,7]. These novel organisms have been successfully identified due to advancements in methods utilizing molecular biology and phylogenetic analysis.

*Kodamaea ohmeri* (homotypic synonym: *Yamadazyma ohmeri*/*Pichia ohmeri*) is a ubiquitous yeast strain frequently employed in the food industry to facilitate the fermentation process of different vegetables and fruits [8]. Presently, this fungus is regarded as a rare, emerging pathogen. It was originally documented as a cause of fungemia in 1994 [9], and it has subsequently been increasingly reported as the cause of human infections [3,10,11].

This yeast has caused concerns regarding its potential impact on public health and its significance in epidemiology worldwide, primarily due to its high mortality rate (40% to 50%) [10,12,13], its propensity to cause outbreaks within healthcare facilities, causing infections in various anatomical sites [12,14,15], and its resistance to certain antifungal medications, such as echinocandins and fluconazole, as evidenced by high Minimum Inhibitory Concentrations (MICs) [10].

The accurate geographic distribution of this microorganism as a causative agent of human infections remains uncertain. This yeast can be mistakenly identified phenotypically as *Candida tropicalis*, *C. albicans*, and *C. glabrata* using conventional laboratory tests, which implies that the number of cases may have been underestimated [12,13]. Moreover, there is a substantial lack of data regarding the virulence and drug resistance of this emerging pathogen, requiring the gathering of additional information to enhance our comprehension. To improve our understanding of the epidemiology and diagnosis of this emerging pathogen, this study aimed to provide a comprehensive description of three *K. ohmeri* isolates responsible for fungemia cases in Honduras as part of an ongoing epidemiological surveillance program. As far as we know, this is the first report of *K. ohmeri* causing fungemia in Honduras.

## 2. Materials and Methods

### 2.1. Collection of Isolates

As a component of a surveillance program in Honduras, tertiary care laboratories under the National Health System were instructed to send all *Candida* isolates from sterile anatomical sites to the National Surveillance Laboratory (NSL) of the Ministry of Health. The purpose was to verify the identification of the *Candida* species and assess their susceptibility to antifungal drugs.

### 2.2. Identification of Isolates through PCR-RFLP of the Ribosomal ITS Region

Seventy-four yeasts were cultured individually in 4 mL of YPD broth (yeast potato dextrose) at 30 °C for 24 h with constant shaking at 200 rpm. The DNA of all isolates was extracted according to a previously published protocol [16]. The amplifications were performed under the following conditions in a volume of 50 μL: 25 of PCR Master Mix (Promega Corporation, Madison, WI, USA), 1 μL of each primer, ITS1 and ITS4, 5′-TCC GTA GGT GAA CCT GCG G-3/5′-TCC TCC GCT TAT TGA TAT GC-3′, and 1 μL of DNA (40 ng/μL). Reactions were carried out with an initial denaturation step at 95 °C for 5 min, 37 cycles at 95 °C for 30 s, 56 °C for 30 s, and 72 °C for 30 s, with a final extension at 72 °C for 5 min. Amplicons were separated by 1.5% agarose gel electrophoresis with ethidium bromide. Four microliters of each PCR product were digested for 2 h at 37 °C with 2 μL of buffer, 0.2 μL of acetylated BSA (10 μg/μL), and 0.5 μL of the restriction enzyme MspI (10 U/μL) (Promega Corporation, Madison, WI, USA). The restriction fragments were analyzed on a 2% agarose gel.

### 2.3. Identification of Candida auris

Isolates that produced bands of around 400 base pairs using primers ITS1 and ITS4, which is in line with the predicted pattern for the *C. haemulonii* complex and its related species *C. auris* [17], were chosen for a targeted PCR assay based on the glycosylphosphatidylinositol (GPI) gene for the diagnosis of *C. auris* [18]. A band size of 137 bp indicated *C. auris*, and no amplification product was expected for the rest of the *Candida* species. Isolates that presented PCR products of around 400 bp for the ITS marker but failed to amplify the GPI gene, which is critical to confirming the presence of *C. auris*, were selected for sequencing.

### 2.4. Sequence Analysis and Construction of Cladograms

The PCR products were purified and sequenced in both directions according to the protocols of the Psomagen company (https://lims.psomagen.com/, accessed on 21 January 2024). The quality of the sequences was analyzed with Geneious prime^®^ 2023.1.2 software, and international databases contained in NCBI were queried to confirm the identity of the sequences using the BLAST tool. The sequences were compared to sequences found in GenBank, and the result with the highest percentage of similarity was recorded as the most likely identification for each isolate. The sequences were submitted to GenBank, and accession numbers were assigned to each strain.

More than 110 homologous sequences from the ITS region of *K. ohmeri* isolated from 17 countries in 4 continents were downloaded and aligned with the sequences obtained in this study. The ClustalW tool of the Geneious software was used to align sequences with the same length. The Tamura–Nei genetic distance model and the Neighbor-Joining method, with a bootstrap of 1000 replicates, were used to construct a cladogram. Homologous sequences from *Candida albicans* and *Schizosaccharomyces japonicus* were also downloaded and included in the alignment as an outgroup.

### 2.5. Phenotypic Identification

#### Microbiological Identification and Chromogenic Culture Media

The isolates were cultured on Sabouraud dextrose agar (SDA, Becton Dickinson, Franklin Lakes, NJ, USA) at 30 °C for 48 h and subsequently cultured in three chromogenic media for the identification of yeasts. CHROMagar (CHROMagar Candida^TM^, Paris, France), CHROMagar^TM^ (Becton, Dickinson and Company, Franklin Lakes, NJ, USA), and Chromatic^TM^ (Candida, Liofilchem^®^, Teramo, Italy) were used. These media were incubated at 37 °C for 48 h and evaluated based on the color of the colonies according to the manufacturers’ instructions. Furthermore, all isolates were identified using the systems BD Phoenix^TM^ (Becton, Dickinson and Company, Franklin Lakes, NJ, USA) and VITEK^®^ 2 (bioMerieux, Craponne, France) following the manufacturers’ instructions.

Also, morphological characteristics, such as chlamydospore formation, were evaluated in corn flour agar and germ tube production or early filamentation following the protocols proposed by Giusiano et al., 2016 [19]. Both tests allow for the presumptive identification of *Candida albicans* complex. *Candida albicans* ATCC 10231 and *C. parapsilosis* ATCC 22019 were used as positive and negative controls, respectively.

### 2.6. Antifungal Susceptibility Testing

The antifungal susceptibility profile of the isolates was determined using the Sensititre^TM^ Yeast-One^TM^ system (Thermo Fisher Scientific, Waltham, MA, USA). Sensititre^TM^ Yeast-One^TM^ is a commercial microdilution method used to determine the Minimum Inhibitory Concentrations (MICs) of yeasts and some filamentous fungi to the following antifungals: anidulafungin (0.015–8 mg/L), micafungin (0.008–8 mg/L), caspofungin (0.008–8 mg/L), fluconazole (0.12–256 mg/L), posaconazole (0.008–8 mg/L), voriconazole (0.008–8 mg/L), itraconazole (0.015–16 mg/L), amphotericin B (0.12–8 mg/L), and 5-fluorocytosine (0.06–64 mg/L). Antifungal susceptibility testing was performed following the manufacturer’s recommendations. Briefly, all isolates were cultured in SDA at 37 °C for 24 h. A suspension was prepared by mixing isolated colonies with sterile, distilled water until it reached a density equivalent to the McFarland Nº 0.5 standard. The accuracy of the density was confirmed using a nephelometer. *C. parapsilosis* ATCC 22019 was used as a reference strain for quality control. The MIC values were evaluated after 24 h of incubation at 37 °C. The yeast’s growth was demonstrated by a transition in color from blue (negative, indicating absence of growth) to red (positive, indicating growth).

### 2.7. Assays for Hydrolytic Enzyme Activity of Kodamaea ohmeri

The phospholipase and hemolytic activities of the *K. ohmeri* isolates were assessed using the established techniques described in earlier publications [20]. *C. albicans* ATCC 10231 was used as a positive control for phospholipase activity and *C. parapsilosis* ATCC 22019 was used as a positive control for hemolytic activity. Proteinase activity was assessed using the determination of caseinase and gelatinase enzymes, as well as the hydrolysis of bovine serum albumin (BSA) [20]. *C. parapsilosis* ATCC 22019 and *C. albicans* ATCC 10231 were used as positive controls for protease activity. The enzymatic activity was evaluated using the enzymatic activity coefficient (Pz). Pz was calculated by dividing the diameter of the colony (A) by the sum of the colony diameter and the hydrolysis/precipitation zone (B), [Pz = A/B] [21]. The isolates were classified according to the Pz value into four categories: Pz = 1.0 no enzymatic activity, Pz = 0.99 to 0.90 weak enzymatic activity, Pz = 0.89 to 0.70 moderate activity, and Pz ≤ to 0.69 strong activity. The Pz values were calculated by taking the average of two independent tests, with each experiment being conducted twice.

### 2.8. Assessment of Kodamaea ohmeri’s Ability to Produce Biofilms

Biofilm formation was determined according to the protocol previously published by Saiprom et al., 2023 [22], with modifications. Briefly, the isolates were cultured in a YPD medium overnight at 37 °C. Afterwards, a yeast suspension was prepared in YPD medium with a density corresponding to the McFarland Nº 0.5 standard. A volume of 100 μL of the suspension was distributed into a 96-well flat-bottom plate and underwent incubation at 30 °C for 48 h. The wells were rinsed thrice with a 1× phosphate buffer solution (PBS) to eliminate cells that were not adhered. The empty wells were allowed to dry for 45 min, and 200 μL of 0.1% crystal violet was added to each well and incubated for 45 min at room temperature. After that, the crystal violet was eliminated, and the plate was left to dry for 10 min at room temperature. Subsequently, the wells were gently rinsed twice with 200 μL of sterile, distilled water. A volume of 300 μL of 100% ethanol was added to remove color from the biofilm, and the plate was kept at room temperature for 45 min. A 150 μL volume of eluted crystal violet was transferred to a new 96-well plate, and the optical density (OD) was measured at 590 nm in a spectrophotometer (Thermo Scientific^TM^ Genesys 20, Oslo, Norway). Sterile YPD was used as a negative control. Two independent experiments were performed for each isolate, and each experiment was repeated three times. The biomass of each isolate was calculated as the average OD value based on the two independent experiments. The production capacity of biofilm was calculated by averaging the OD of each sample (ODs). The OD of the negative control (ODnc) was also determined. The strains were classified using the following criteria: non-producer (ODs ≤ ODnc); weak producer (ODnc < ODs ≤ 2 × ODnc); moderate producer (2 × ODnc < ODs ≤ 4 × ODnc); and strong producer (ODs > 4 × ODnc) [23]. *C. albicans* ATCC 10231 was used as a positive control of biofilm production.

### 2.9. Invasion and Adherence Assays

The invasion assays were carried out according to the methodology proposed by Faria-Gonçalves et al., 2022 and Saiprom et al., 2023 [22,24], with modifications. Briefly, RAW 264.7 macrophages were reactivated in culture dishes with Dulbecco’s modified Eagle’s medium (DMEM) (Merck, Darmstadt, Germany) supplemented with 5% fetal bovine serum (FBS) (GIBCO^TM^, Fisher Scientific Inc., Madrid, Spain). The cultured cells were placed in an incubator at 37 °C and 5% CO_2_ until they reached 80% confluency. Subsequently, in six-well polystyrene plates with coverslips, a cell suspension equal to 50,000 cells/mL was prepared in 2 mL of DMEM supplemented with 5% FBS and antibiotics (Penicillin–Streptomycin). Plates were incubated overnight at 37 °C in 5% CO_2_. RAW 264.7 cell monolayers were washed with sterile 1× PBS, and after washing, 2 mL of fresh DMEM supplemented with 5% FBS and antibiotics was added to each well. From a 48 h culture of the three isolates under study, a suspension was made in DMEM containing 5% FBS with a density equivalent to the McFarland Nº 0.5 standard (approximately 1 × 10^6^ CFU/mL). The RAW 264.7 cells were infected with each *K. ohmeri* isolate at a multiplicity of infection (MOI) of 5 and incubated for 4 h at 37 °C in 5% CO_2_. Uninfected cells were used as a negative control. After 4 h, the cultures were washed three times with 1× PBS to remove planktonic yeasts. The cells were fixed with methanol for 15 min, stained with Giemsa for 15 min, and subsequently washed with sterile water. The coverslips were removed from each well and fixed on a slide for visualization through bright-field microscopy at 100× objectives to observe their morphology and detect the invasive yeasts. The cells were counted in 10 microscopic fields (approximately five cells per field), the number of macrophages invaded with yeast was determined, and the percentage of invasion was obtained. Each strain was evaluated in triplicate.

Adherence tests were performed following previously published protocols with modifications [22,24]. For the adhesion assays, the HeLa cell line was used, which was cultured on plates with DMEM medium supplemented with 5% FBS, which was incubated at 37 °C in 5% CO_2_ until 80% confluency was observed. Subsequently, in six-well polystyrene plates with coverslips, a cell suspension of 50,000 cells/mL was prepared in 2 mL of DMEM supplemented with 5% FBS and antibiotic (Penicillin–Streptomycin). Plates were incubated for 24 h at 37 °C in 5% CO_2_. HeLa cell monolayers were washed with sterile 1× PBS, and after washing, 2 mL of fresh DMEM was added to each well. From a 48 h culture of the yeast under study, and in DMEM culture medium, a cell suspension equivalent to the McFarland standard Nº 0.5 (approximately 1 × 10^6^ CFU/mL) was made.

HeLa cells were infected with each *K. ohmeri* isolate at a multiplicity of infection (MOI) of 10 and incubated for 2 h at 37 °C in 5% CO_2_. Uninfected cells were implemented as a negative control. After 4 h, cultures were washed three times with 1× PBS to remove non-adherent yeast, and adhered cells were fixed with methanol for 15 min. To visualize the adhered cells, Giemsa staining was performed, followed by three washes with 1× PBS, and, finally, they were observed under the microscope using the 40× and 100× objectives. The HeLa cells present per field were counted in 10 fields (approximately 200 HeLa per field), the yeasts adhered to the HeLa cells, and the number of cells with adherence was quantified. The results are expressed as the percentage of cells with adhered yeasts and the number of adhered yeasts per HeLa cell. Each isolate was analyzed in independent experiments in triplicate.

For invasion and adherence assays, standard deviation (SD) was calculated by counting a total of 10 fields per slide. A statistical analysis was performed using one-tailed ANOVA and Tukey’s multiple comparisons test.

## 3. Results

A total of seventy-four yeast isolates, which were phenotypically identified as *Candida* spp. and obtained from blood cultures, were submitted to the Laboratory of National Health Surveillance (LNS) of the Ministry of Health for molecular identification and assessment of their susceptibility to antifungal drugs. Three isolates, identified as C-29, C-35, and C-38, were initially believed to be *Candida auris*. These isolates were obtained from two public hospitals in San Pedro Sula (n = 2) and Tegucigalpa (n = 1).

The molecular identification of the three yeast isolates was conducted using the PCR-RFLP technique targeting the ribosomal ITS region and the MspI enzyme. This approach is commonly employed for the identification of *Candida* species and complexes [25,26,27,28]. The PCR products from the three isolates had a length of around 400 base pairs, and no cleavage occurred when the products underwent digestion with the MspI enzyme (Appendix A). The amplified size obtained using the ITS1 and ITS4 primers matched the predicted size for the species grouping in the *C. haemulonii* complex (*C. haemulonii* s.s., *C. haemulonii* var. *vulnera*, *C. duobushaemulonii*, *C. pseudohaemulonii*) and its related species *C. auris* [17]. Consequently, a targeted PCR was conducted to detect *C. auris* using the GPI gene, but no amplification result was detected in the three isolates (Appendix A). Subsequently, the ITS region was amplified and sequenced [17], the resulting sequences of the three isolates were edited and analyzed using the NCBI BLAST tool, and all isolates were identified as *Kodamaea ohmeri*. The sequences obtained were deposited in GenBank under accession numbers OR791936–OR791938. The nucleotide sequencing of the three isolates showed no discrepancies. A phylogenetic tree was created using the ITS sequences of this study and by comparing them with sequences of *K. ohmeri* isolated from different geographical locations. The three sequences obtained in this study were not identical to any single nucleotide polymorphisms (SNPs) (Figure 1). A phylogenetic tree was constructed by aligning the ITS sequences obtained from this study with more than 110 sequences of *K. ohmeri* isolated from various geographical areas. The *K. ohmeri* isolates from Honduras exhibited a high degree of similarity, forming a cohesive cluster with most sequences. No clear phylogenetic clustering was seen based on the geographic origin of the isolates (Figure 1).

In this work, we assessed three isolates of *K. ohmeri* using phenotypic identification methods. This was because there is insufficient information available on clinical isolates of *K. ohmeri* and their identification profile in routine laboratory conditions. The three isolates cultivated in CHROMagar^TM^ Candida, CHROMagar^TM^, and Chromatic^TM^ Candida produced colonies with an iridescent-pink appearance within the initial 18 to 24 h, which then became green after 48 h of incubation in both media (Figure 2).

Similarly, the BD Phoenix^TM^ and VITEK^®^ 2 automated identification systems were used to analyze these isolates. The VITEK^®^ 2 system successfully identified the isolates as *K. ohmeri*, with identity percentages of 96% or higher. However, the BD Phoenix^TM^ system failed to identify two isolates, and one of them was mistakenly identified as *C. albicans* with an identity percentage of 95%.

Phenotypic tests were conducted to examine the morphology of the isolates. Specifically, the creation of chlamydospores in corn flour agar and the development of germ tubes were assessed. However, no similarity to *C. albicans* was seen in either of these tests (i.e., no chlamydospores or germ tubes were produced). The MIC results of the three isolates for nine antifungals are shown in Table 1. The MIC values for voriconazole, posaconazole, and itraconazole varied between 0.03 and 0.12 mg/L. However, fluconazole exhibited an MIC of 4 mg/L in all isolates. The MIC range for the echinocandin family ranged from 0.12 to 2 mg/L, with the highest value recorded in an isolate for caspofungin. The MIC range for amphotericin B was between 0.25 and 0.5 mg/L, whereas all isolates showed an MIC of ≤0.06 mg/L for 5-fluorocytosine.

The three isolates exhibited biofilm generation and extracellular enzyme synthesis after 48 h of incubation at 37 °C. The absorbance values ranged from 0.727 to 1.245. Based on the measured absorbances, all isolates demonstrated a high level of biofilm production. In addition, all isolates of *K. ohmeri* had a Pz value ≤ 0.69 for the synthesis of extracellular enzymes, indicating significant hemolytic, phospholipase, and protease (gelatin, casein, BSA) activity. The absorbances and Pz values are shown in Table 2.

In addition, the capacity of isolates to adhere to and invade cells was assessed using the HeLa and RAW 264.7 cell monolayer infection models (Figure 3). All isolates exhibited adhesion to HeLa cells, whereas C-29 and C-38 demonstrated the highest level of adherence. There was a statistically significant difference in the number of HeLa cells with yeast in strain C-29 compared to the other two (*p* < 0.05). Each isolate had a minimum of two pseudohyphae per HeLa cell. Similarly, the rate of invasion in RAW 264.7 cells ranged from 30% to 35%. Figure 3 displays the outcomes of adhesion in HeLa cells and invasion in RAW 264.7. All isolates of *K. ohmeri* exhibited a significant ability to form filaments in both experiments.

## 4. Discussion

Fungal infections pose a significant public health concern. Approximately 1.5 million individuals perish each year due to fungal infections, while a much larger number have serious fungal diseases [29]. The prospect of fungal infections appears unfavorable in the foreseeable future due to several factors. These include the rising number of individuals vulnerable to fungal infections, the growing prevalence of resistance to various antifungal medications, limited accessibility to treatment options, and the restricted availability of methods for detecting and identifying fungi. Additionally, there is a concerning emergence of new human fungal pathogens that possess inherent resistance to currently available antifungal drugs [4,5,7,30,31].

Environmental and human-induced factors are greatly impacting the evolution, selection, and adaptation of previously harmless fungal species, which are now posing a growing threat to the health of humans and animals [32]. This is the case for *Candida viswanathii*, *C. auris*, *C. blankii*, *C. palmioleophila*, *C. vulturna*, *C. massiliensis*, *Trichophyton indotineae*, *Pseudogymnoascus destructans*, *Batrachochytrium dendrobatidis*, and *Kodamaea ohmeri*, all of which are environmental fungi that have been successfully adapted to animals or humans over time [12,18,31,33,34,35,36].

The genus *Kodamaea* consists of six species: *K. anthophila*, *K. kakaduensis*, *K. laetipori*, *K. nitidulidarum*, *K. samutsakhonensis*, and *K. ohmeri*. However, only *K. ohmeri* has been proven to have clinical significance [8]. *K. ohmeri* is commonly associated with insects and flowers [8]; however, it is now considered a rare, emerging pathogen. In recent years, its significance has increased due to the decreased effectiveness of certain antifungal medications, like echinocandins and azoles. Additionally, it is associated with a mortality rate of approximately 50% [12,13].

Since its initial report of causing fungemia in a 64-year-old male patient in 1994, cases of human infections attributed to this yeast have been on the rise in multiple nations across Asia, America, Europe, and Africa [10,12,37]. While China and India report the largest number of cases, *K. ohmeri* has also been documented in the Americas, including the USA (n = 7), Brazil (n = 3), Colombia (n = 1), and Mexico (n = 1) [12]. A national surveillance program in Honduras has detected three isolates of *K. ohmeri* in the cities of San Pedro Sula and Tegucigalpa in response to recent reports of emergent yeasts worldwide. Initially, the three isolates were misidentified as *Candida albicans* due to their morphological traits. This finding is highly significant in the local context as it emphasizes the importance of implementing an ongoing surveillance program that utilizes molecular tests and/or mass spectrometry to accurately identify filamentous fungi and yeasts [12].

*K. ohmeri* isolates were cultivated on three commercially available chromogenic agars specifically formulated for yeast species identification based on colony color. After 48 h, the three isolates exhibited green colonies, which might have led to a misidentification as *C. albicans*. Agrawal et al., 2014 described different morphotypes of *K. ohmeri* in HiCrome^TM^ Candida, where the colors of the colonies showed pink during the first 24 h, and then turned green or blue at 48 h until evolving to metallic blue at 72 h [38]. In their 2007 study, Lee et al., reported the growth of pink and green colonies in CHROMagar Candida^TM^, which transitioned into blue colonies after 72 h [39]. Similar results were reported by Biswa et al., 2015, Yu et al., 2019, and Mtibaa et al., 2019 in this medium [37,40,41]. In this sense, it seems that chromogenic agars play an important role in the identification of the most important species of the genus *Candida* (*C. albicans*, *C. tropicalis*, *C. parapsilosis*, and *C. krusei*), but they do not represent a viable approach for the identification of emerging yeasts.

*K. ohmeri* is frequently misdiagnosed as *C. tropicalis*, *C. glabrata*, or *C. albicans* during routine testing [13,39,42], which would have an impact on under-reporting of clinical cases due to this species. This study also utilized traditional phenotypic assays for assessing *K. ohmeri* isolates. None of the three strains exhibited the ability to generate either a germ tube or chlamydoconidia on corn flour agar. Consequently, these techniques can differentiate between species within the *C. albicans* complex, but they lack sufficiency in identifying additional *Candida* species or those belonging to other genera of yeast. Regarding automated identification methods, VITEK^®^ 2 demonstrated successful identification of the three isolates. This is not surprising, as this system can identify up to 50 different yeasts, including *K. ohmeri*. However, it has been observed that in this identification system, certain isolates of *Candida palmioleophila* can be mistakenly identified as *K. ohmeri* [33]. The BD Phoenix BD Phoenix^TM^ yeast identification system, on the other hand, could not correctly identify any isolate. This is in line with what the manufacturer states, because this equipment can identify up to 64 species of yeasts, but not the *Kodamaea* species.

Sequencing of ITS regions has been recognized as a reference method for the characterization of yeasts, including *K. ohmeri* [10,12]. After comparing the sequences obtained in this study with over 110 ITS sequences of clinical and environmental isolates of *K. ohmeri* from different regions worldwide, it became clear that they formed a monophyletic group without any cluster formation. Our findings, along with the existing genomic data, suggest that the genetic diversity of this emerging species is restricted. In contrast, one study used a Multi-Locus Sequence Typing (MLST) approach with six *K. ohmeri* isolates causing infections in Bangladesh, and the authors reported five different allelic profiles [43]. The authors present a dendrogram displaying eight distinct clusters. This dendrogram was created using ITS sequences from their six isolates, as well as homologous sequences from various geographical locations. However, it is important to note that the sequences labeled under accession numbers OQ606981–OQ606985 do not exhibit any variations among themselves. Consequently, these findings contradict our results.

Concerning the susceptibility profiles of the antifungal drugs, the isolates exhibited low MIC for amphotericin B, triazoles, 5-flucytosine, anidulafungin, and micafungin. However, the MIC values for caspofungin were found to be high. These results are consistent with the evidence collected in two recent reviews [10,12]. While there are no accepted epidemiological cutoff values or clinical breakpoints for *K. ohmeri* according to EUCAST [44] or CLSI [45], it is crucial to determine the susceptibility profile to antifungals in emerging yeasts. This information assists the clinician in determining the most suitable course of therapy. It is imperative to allocate resources toward incorporating antifungal susceptibility testing into routine laboratories, especially in low- and middle-income countries where access to these tests is limited [12]. We hope that our data advance the epidemiological understanding of the susceptibility profile of *K. ohmeri*.

Yeast pathogenesis seems to be significantly influenced by the production of biofilms and the presence of extracellular hydrolytic enzymes [46,47]. The isolates we tested exhibited strong biofilm production, which aligns with the findings of Giacabino et al., (2015). In their study, they investigated yeasts linked with fungal peritonitis and identified one of the isolates as *K. ohmeri*. This strain exhibited a high biofilm production capability, ranking as the second most proficient isolate in biofilm production [48]. Similarly, our findings align with the data presented by Maciel et al., 2019, who examined potentially harmful yeasts obtained from beaches in Brazil. These authors assessed five strains of *K. ohmeri*, which exhibited the maximum biofilm production among all the strains examined [49]. Several fungi of the phylum Ascomycota have been shown to form biofilms. Among yeasts, *Candida* species have been extensively investigated as one of the main models [50,51]. Biofilms affect the long-term survival of microbes on non-living surfaces and tissues. They provide protection against host immune cells and are strongly linked to resistance to antifungal treatments [52]. Further research is required to establish the correlation between the development of *K. ohmeri* biofilm and its resistance to antifungal agents, as well as its ability to persist on non-living surfaces.

The isolates had a high potential for producing extracellular enzymes, specifically proteases, phospholipase, and hemolysin. Hydrolytic enzymes have an important role in adhering to and infiltrating host cells [20,22,46]. Proteases facilitate the entry into cells by breaking down proteins like albumin, collagen, and mucin. They also contribute to the breakdown of antibodies, complement factors, and cytokines [20,47]. Phospholipases alter the cell membrane, whereas hemolysins assist in the destruction of cells and the uptake of iron [20,46,47].

This study also aimed to assess the adhesive potential of three *K. ohmeri* isolates to the HeLa cell line. This methodology has previously been employed to demonstrate the capacity of several *Candida* species to adhere [24,53,54,55]. For yeasts to successfully colonize and infiltrate tissues, they need to adhere themselves to the epithelium. This attachment is facilitated by adhesins, integrins, and cadherins, which are located on the surface of the fungal cell wall [53,56]. After adhering, yeasts can penetrate host cells either through induced endocytosis or active penetration [53,57]. Our results demonstrate a significant index of *K. ohmeri*’s adherence to this specific cell line. Furthermore, all three isolates exhibited notable morphological transformations from yeast to pseudohyphae, indicating that these modifications may facilitate host invasion through active penetration. Further research is required to investigate the adhesion processes of *K. ohmeri* in more depth.

In addition, we demonstrated the effectiveness of the RAW 264.7 macrophage line in effectively engulfing all three strains of *K. ohmeri*. Macrophages are cells specialized in the detection, phagocytosis, and destruction of fungal propagules [58,59], that are recognized through receptors that interact with molecular patterns associated with pathogens, which are generally arranged along the cell surface of fungal cells [58]. Mannans, galactomannans, glucosylceramides, mannoproteins, chitin, melanin, and β-glucans are key molecules that interact with and activate the immune system [58,59]. Our results highlight the need to elucidate the molecular mechanisms by which immune cells interact with *K. ohmeri*. However, this study solely assessed the extent of cellular internalization without examining the lethal activity of macrophages. Therefore, further investigations are required to ascertain and comprehend the survival rate and the processes via which yeasts interact with macrophage phagosomes. This information would enhance our comprehension of potential strategies for escape and immune recognition, while also aiding in the identification of novel treatment targets for this emerging pathogen.

## 5. Conclusions

To our knowledge, this is the first report of *K. ohmeri* in Honduras causing human infection. Our findings highlight the importance of strengthening the capacities of mycology reference laboratories and epidemiological surveillance in the country. On the other hand, our results demonstrate that the identification of emerging pathogens, such as *K. ohmeri*, is a challenge for clinical laboratories and that the implementation of tools based on molecular biology and/or mass spectrometry is urgently needed in the main hospital centers of the main cities of the country because some pathogens could be misdiagnosed. The implementation of these tools would contribute to decision making for the containment of outbreaks and the spread of emerging fungal pathogens. Furthermore, here we present important information about some characteristics that could be mediating the virulence of *K. ohmeri*.

## 6. Limitations

The patients’ medical records, with key data on risk factors, underlying diseases, and treatment procedures, were inaccessible. Sensititre^TM^ Yeast-One^TM^ system is a strongly recommended approach for assessing the susceptibility of yeasts, especially *Candida* species, to antifungal drugs [60,61,62]; however, Zhou et al., (2019) found that for uncommon yeasts, like *K. ohmeri*, the MIC values may vary due to the absence of standardized tests. Therefore, it is advisable to employ the broth microdilution method, which offers greater reliability and precision [12]. Due to the unavailability of broth microdilution in Honduras, it was not feasible to estimate the Minimum Inhibitory Concentrations (MICs) using this method.

## Figures and Tables

**Figure 1 jof-10-00186-f001:**
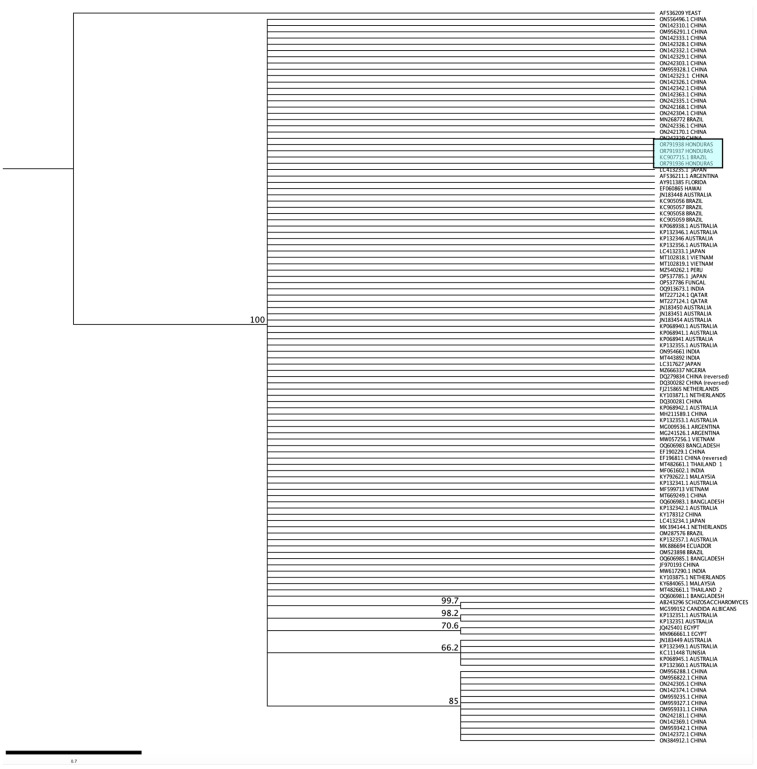
Phylogenetic analysis of *K. ohmeri* based on the internal transcribed spacer (ITS) regions of the 18S rDNA gene of isolates from different geographical regions. The Tamura–Nei genetic distance model and the Neighbor-Joining method, with a bootstrap of 1000 replicates, were used to construct a cladogram. The sequences obtained in this study are shown inside the light blue box.

**Figure 2 jof-10-00186-f002:**
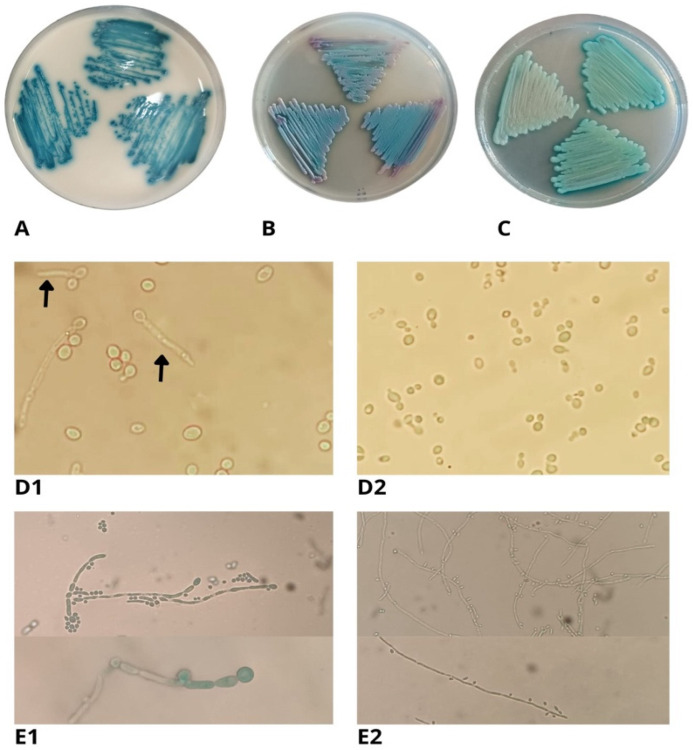
Growth of *Kodamaea ohmeri* in (**A**) CHROMagar medium (CHROMagar Candida^TM^, Paris, France), (**B**) CHROMagar^TM^ (Becton, Dickinson and Company, Franklin Lakes, NJ, USA), (**C**) Chromatic^TM^ Candida, Liofilchem^®^. (**D1**) Germ tube production (arrows) of the positive control (*C. albicans* ATCC 10231). (**D2**) Germ tube test in *K. ohmeri* (non-producer). (**E1**,**E2**) Photomicrographs with 40× objective of fungal structures obtained from culture on corn flour agar at 30 °C for 48 h of (**E1**) the positive control (*C. albicans* ATCC 10231) and (**E2**) *K. ohmeri* showing pseudohyphae with ovoid and elongated blastoconidia on the sides.

**Figure 3 jof-10-00186-f003:**
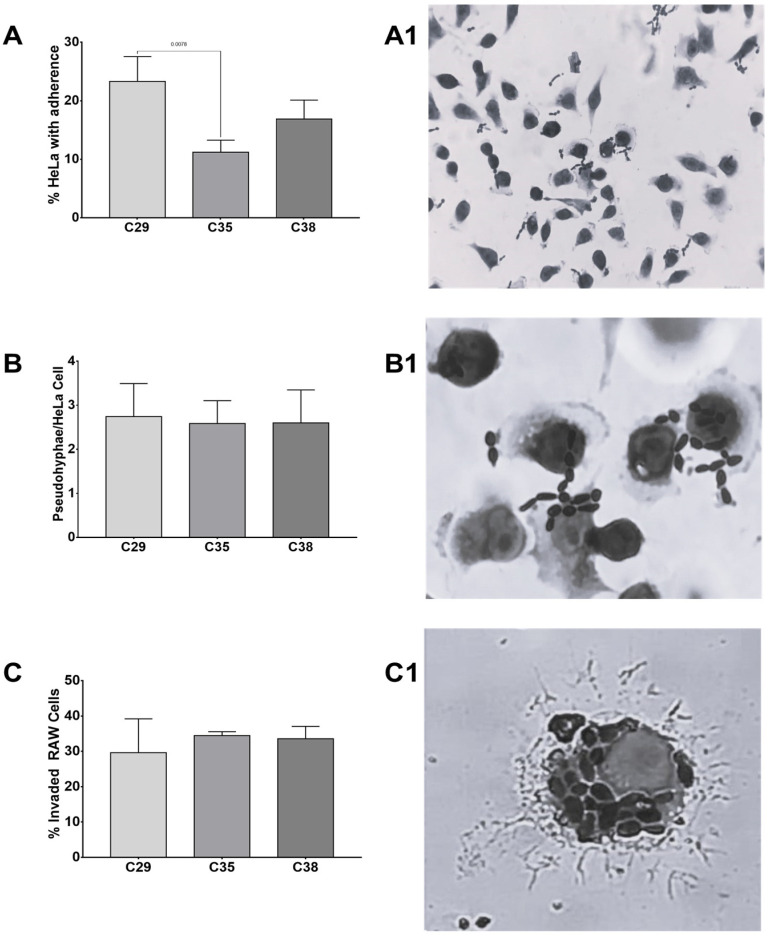
Adhesion and invasion assays in HeLa cells. (**A**) Percentage of HeLa cells that exhibited adherent yeasts; (**A1**) Micrograph obtained through brightfield microscopy with 40× objective. Giemsa stain showing yeasts and pseudohyphae of strain C-29 adhered to HeLa cells; (**B**) Number of pseudohyphae per HeLa cell; (**B1**) Micrograph obtained through brightfield microscopy at 100× objective. Giemsa stain. Pseudohyphae adhered to HeLa cells; (**C**) Percentage of macrophages per field invaded by strain; (**C1**) Micrograph obtained through brightfield microscopy at 100× objective. Giemsa stain. Macrophage invaded by blastoconidia of strain C-29.

**Table 1 jof-10-00186-t001:** Evaluation of the Minimum Inhibitory Concentration (MIC) of nine antifungal drugs on three isolates of *Kodamaea ohmeri* using the Sensititre^TM^ Yeast-One^TM^ system.

Antifungal Drug	Isolate (MIC)		
	C-29 (mg/L)	C-35 (mg/L)	C-38 (mg/L)
Amphotericin B	0.5	0.25	0.5
Anidulafungin	0.25	0.12	0.12
Caspofungin	2.0	1.0	0.25
Micafungin	0.12	0.25	0.12
Fluconazole	4.0	4.0	4.0
Itraconazole	0.12	0.12	0.12
Posaconazole	0.06	0.12	0.06
Voriconazole	0.03	0.03	0.03
5-Flucytosine	≤0.06	≤0.06	≤0.06

**Table 2 jof-10-00186-t002:** Analysis of biofilm production and coefficient of extracellular enzymatic activity (Pz).

		(Pz)				
Isolate	Biofilm Production	Gelatin	BSA	Casein	Phospholipase	Hemolisin
C-29	0.831 ± 0.12 Strong producer	0.57 ± 0.04 Strong activity	0.48 ± 0.02 Strong activity	0.83 ± 0.06 Moderate activity	0.67 ± 0.03 Strong activity	0.30 ± 0.02 Strong activity
C-35	1.245 ± 0.16 Strong producer	0.51 ± 0.05 Strong activity	0.38 ± 0.05 Strong activity	0.89 ± 0.01 Moderate activity	0.62 ± 0.03 Strong activity	0.30 ± 0.04 Strong activity
C-38	0.727 ± 0.15 Strong producer	0.56 ± 0.02 Strong activity	0.46 ± 0.04 Strong activity	0.89 ± 0.02 Moderate activity	0.67 ± 0.05 Strong activity	0.30 ± 0.02 Strong activity

## Data Availability

Data are contained within the article and Appendix A.

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
