# Peer review of "First Report of the Emerging Pathogen Kodamaea ohmeri in Honduras"

_jof, 2024, doi:10.3390/jof10030186_

Round 1

Reviewer 1 Report

The study describes the virulence factors, antifungal susceptibility, and genetic correlation of three isolates of K. ohmeri obtained from bloodstream samples in Honduras. Given a lack of available data regarding the features of this species, the study expands the knowledge associated with geographic distribution, virulence, and resistance in this rare opportunistic species. The relevance is that K. ohmeri can be identified phenotypically as other common pathogenic yeast species such as Candida tropicalis, C. albicans, and C. glabrata by conventional laboratory tests, and authors have addressed this issue properly. However, the reader may lose focus with the way the methodology was written. The item is too long and would benefit from a significant focus on the relevant points to identify K. ohmeri. The procedures and results for a surveillance program in Honduras are quite interesting for another article but are out of the study scope. I suggest only informing the frequency of K. ohmeri in such surveillance. In this context: What is the rationale for searching and identifying Candida auris in this study? Moreover, why do the authors use tests for the presumptive identification of Candida albicans complex? Finally, the authors should pay attention to discussing only the relevance of results focusing on K. ohmeri.

Minor comments:

Inform the number of strains at the beginning of the Methodology section.

Revise the use of italics in Latin names throughout the text.

Line 37. Use the acronym MIC for Minimum Inhibitory Concentration

Line 53 Use MIC for Minimum Inhibitory Concentration and avoid plural for acronyms. Instead of MICs, prefer MIC values or MIC results

Line 74. Define YPD

Line 91. Define GPI

Line 126. BD Phoenix is in duplicate.

Line 134. Prefer Antifungal susceptibility testing, instead of Antifungal sensitivity testing. The same instruction for antifungal susceptibility instead of antifungal sensitivity

Line 168. Define PBS

Line 182. Use lowercase for non.

Lines 290-292. Delete these lines since there is no relevant

Line 295. The idea “ …all isolates showed a MIC of 4 mg/L”  is incorrect. Minimum Inhibitory Concentration is related to drug and not to the isolate.

Lines 393-395 These sentences are irrelevant, please delete them.

Line 410. Define MLST

Line 418. The use of capital letter for 5-flucytosine is incorrect.

Line 420. Inform the results collected in the two cited reviews.

Line 424. Please, clarify the idea in Furthermore, the calculation of MICs has previously contributed to the identification of novel or emerging pathogens

Lines 393-395 These sentences are irrelevant, please delete them.

Line 410. Define MLST

Line 418. The use of capital letter for 5-flucytosine is incorrect.

Line 420. Inform the results collected in the two cited reviews.

Line 424. Please, clarify the idea in Furthermore, the calculation of MICs has previously contributed to the identification of novel or emerging pathogens

Figure 2. It is unnecessary to depict: E) Photomicrograph with 40× objective of fungal structures 285 obtained from culture on corn flour agar at 30 °C, for 48 h of (E1) the positive control (C. albicans 286 ATCC 10231), and (E2) K. ohmeri showing pseudohyphae with ovoid and elongated blastoconidia 287 on the sides since it does not provide relevant information

Figure 3. The term yeasts is not appropriate, but blastoconidia

Table 2. The enzymatic activity lacks interpretation for each isolate. BSA legend is missing.  

Figures in Supplemental Material: lacks numbers in several lanes  

Author Response

February 19th, 2024

Editor

Journal of Fungi

MDPI

Dear Editor:

We want to express our deepest gratitude for the revisions made to the manuscript entitled "First report of the emerging pathogen Kodamaea ohmeri in Honduras". We have attended to the observations of both reviewers, and we have tried to respond point by point to each of them. Furthermore, we have made the necessary changes to the manuscript, and we trust that the new version will satisfy both the editor and the reviewers.

The changes to the document have been highlighted in yellow, and below you will find responses to your comments.

Thanking you

Sincerely,

Gustavo Fontecha, PhD

Reviewer 1

  1. The study describes the virulence factors, antifungal susceptibility, and genetic correlation of three isolates of  ohmeriobtained from bloodstream samples in Honduras. Given a lack of available data regarding the features of this species, the study expands the knowledge associated with geographic distribution, virulence, and resistance in this rare opportunistic species. The relevance is that K. ohmeri can be identified phenotypically as other common pathogenic yeast species such as Candida tropicalis, C. albicans, and C. glabrata by conventional laboratory tests, and authors have addressed this issue properly. However, the reader may lose focus with the way the methodology was written. The item is too long and would benefit from a significant focus on the relevant points to identify K. ohmeri. The procedures and results for a surveillance program in Honduras are quite interesting for another article but are out of the study scope. I suggest only informing the frequency of K. ohmeri in such surveillance. In this context: What is the rationale for searching and identifying Candida auris in this study? Moreover, why do the authors use tests for the presumptive identification of Candida albicans complex? Finally, the authors should pay attention to discussing only the relevance of results focusing on K. ohmeri.

Answer:

We appreciate the reviewer's observation. The materials and methods section explains the somewhat accidental rationale for discovering the three K. ohmeri isolates. The yeast surveillance program's goal is to identify and confirm Candida species, with a particular emphasis on detecting C. auris, as this emerging pathogen has not been reported in Honduras. Three isolates exhibited a restriction pattern (ITS/PCR RFLP) consistent with C. auris but did not produce amplification with the especific primers for C. auris, prompting the need for sequencing. We have revised the M&M section as per your advice by shortening it and eliminating the description of Cauris's identification. The revised paragraph is as follows: “Identification of Candida auris. Isolates that produced bands of around 400 base pairs using primers ITS1 and ITS4, which is in line with the predicted pattern for the C. haemuloniicomplex and its related species C. auris [17], were chosen for a targeted PCR assay based on the GPI gene for the diagnosis of C. auris [18]. A band size of 137 bp indicated C. auris and no amplification product was expected for the rest of the Candida species. Isolates that presented PCR products of around 400 bp for the ITS marker, but failed to amplify the GPI gene, which is critical to confirming the presence of C. auris, were selected for sequencing”.

Presumptive assays, including corn flour agar, germ tube development, early filamentation, and culture in chromogenic media, were utilized in this work to identify the C. albicans complex. Considering that three isolates of K. ohmeri were first misidentified as C. albicans in hospitals, we found it important to assess how K. ohmeri would react in the tests typically performed on the C. albicans complex in our country. The authors suggest that this information could be pertinent to clinical microbiologists working in laboratories in resource-constrained nations. Concerning the discussion, we have deleted two paragraphs to focus on the epidemiology and relevance of K. ohmeri as an emerging yeast.

Minor comments:

  1. Inform the number of strains at the beginning of the Methodology section.

Answer:

The following sentence has been added: “Seventy-four yeasts were cultured individually.”

  1. Revise the use of italics in Latin names throughout the text.

Answer:

We have reviewed the document exhaustively and have modified three names. We have not modified the term Candida when used as part of a commercial name for a culture medium.

  1. Line 37. Use the acronym MIC for Minimum Inhibitory Concentration

Answer:

Done.

  1. Line 53 Use MIC for Minimum Inhibitory Concentration and avoid plural for acronyms. Instead of MICs, prefer MIC values or MIC results.

Answer:

Done.

  1. Line 74. Define YPD

Answer:

Done.

  1. Line 91. Define e GPI

Answer:

Done: glycosylphosphatidylinositol

  1. Line 126. BD Phoenixis in duplicate.

Answer:

Done. Thanks.

  1. Line 134. Prefer Antifungal susceptibility testing, instead of Antifungal sensitivity testing. The same instruction for antifungal susceptibility instead of antifungal sensitivity

Answer:

Done, thanks.

  1. Line 168. Define 

Answer:

Done. Phosphate buffer solution (PBS).

  1. Line 182. Use lowercase for 

Answer:

Done.

  1. Lines 290-292. Delete these lines since there is no relevant

Answer:

We appreciate the reviewer's observation, however the authors have decided to maintain that information for the reasons described in our first response: On the other hand, in this study presumptive tests were used for the identification of the C. albicans complex, such as corn flour agar and germ tube production or early filamentation, as well as cultivation in chromogenic medium. Considering that the three K. ohmeri isolates were presumptively identified in hospitals as C. albicans, we believed it was relevant to evaluate how K. ohmeri would behave in the analyses to which the C. albicans complex is subjected in our country. The authors believe that this information may be relevant for clinical microbiologists who work in laboratories in countries with limited resources.

  1. Line 295. The idea “ …all isolates showed a MIC of 4 mg/L”is incorrect. Minimum Inhibitory Concentration is related to drug and not to the isolate.

Answer:

The sentence has been modified as follows:However, fluconazole exhibited a MIC of 4 mg/L in all isolates”.

  1. Lines 393-395 These sentences are irrelevant, please delete them.

Answer:

The authors have decided to keep that information for the reasons described in our first response.

Line 410. Define MLST.

Answer: Done. Multi-Locus Sequence Typing (MLST).

  1. Line 418. The use of capital letter for 5-flucytosine is incorrect.

Answer:

Changed.

  1. Line 420. Inform the results collected in the two cited reviews.

Answer:

EUCAST and CLSI are institutions that provide clinical and epidemiological cutoff values for fungal species causing human infections. Due to insufficient Kodamaea isolates, these guidelines do not provide the necessary data for clinical laboratories to determine thresholds that categorize strains as susceptible or resistant.

  1. Line 424. Please, clarify the idea inFurthermore, the calculation of MICs has previously contributed to the identification of novel or emerging pathogens

Answer:

This sentence has been deleted, since it would require a very extensive explanation that would make the discussion even longer.

  1. Figure 2. It is unnecessary to depict: E) Photomicrograph with 40× objective of fungal structures 285 obtained from culture on corn flour agar at 30 °C, for 48 h of (E1) the positive control (C. albicans 286 ATCC 10231), and (E2) K. ohmeri showing pseudohyphae with ovoid and elongated blastoconidia 287 on the sides since it does not provide relevant information

Answer:

It is true that figure 2.E is not essential, however it shows that the Kodamaea isolates do not generate chlamydoconidia, compared to the positive control of C. albicans.

  1. Figure 3. The term yeasts is not appropriate, but blastoconidia

Answer:

The legend has been modified.

  1. Table 2. The enzymatic activity lacks interpretation for each isolate. BSA legend is missing.

Answer:

Table 2 has been modified as follows:

(Pz)

Isolate 

Biofilm production

Gelatin

BSA

Casein 

Phospholipase

Hemolisin

C-29 

0.831 ± 0.12 

Strong producer

0.57 ± 0.04 

Strong activity

0.48 ± 0.02

Strong activity

0.83 ± 0.06 

Moderate activity

0.67 ± 0.03 

Strong activity

0.30 ± 0.02

Strong activity 

C-35 

1.245 ± 0.16 

Strong producer

0.51 ± 0.05 

Strong activity

0.38 ± 0.05 

Strong activity

0.89 ± 0.01 

Moderate activity

0.62 ± 0.03 

Strong activity

0.30 ± 0.04 

Strong activity

C-38 

0.727 ± 0.15 

Strong producer

0.56 ± 0.02 

Strong activity

0.46 ± 0.04 

Strong activity

0.89 ± 0.02 

Moderate activity

0.67 ± 0.05 

Strong activity

0.30 ± 0.02 

Strong activity

  1. Figures in Supplemental Material: lacks numbers in several lanes  

Answer:

The legend of Figure 1 has been improved.

Reviewer 2 Report

The manuscript jof-2876939 entitled “First report of the emerging pathogen Kodamaea ohmeri in Honduras” describe the characteristics of three K. ohmeri isolates from blood specimens. Researchers in the closely related field may be interested in the information provided by this study. 

In addition to only three isolates, another limitation described by authors, the susceptibility testing could be done by broth microdilution method. In the discussion, the authors may add how to improve the detection of emerging species in their national surveillance system.  

There are several concerns as following.

1.     Method: There may be a little confusion for readers at beginning if the paper related to K. ohmeri or C. auris.

2.     With the exception of morphology assay, the authors did not use any controls of different species for other assays. It is not easy for readers to compare the results to known reported.

3.     Figure 2, it would be reader friendly to have scale bars on the pictures such that the readers can appreciate the different size of cells between C. albicans and K. ohmeri. By doing so, it would be clear to know the magnification of the top and bottom panels of E1 and E2.

4.     Figure 2, it would be nice to show photos of all three K. ohmeri isolates.

5.     Figure 2, E1 of morphology of C. albicans seems like pseudohyphal form instead of true hyphal one.

6.     If it is possible, please label the isolates with the contributing hospitals. That is, which one was from one hospital and the remaining two were from the other hospitals.

7.     Line 248, please italicize C. auris.

8.     Line 258, please describe as “identical” instead of “100% similarity”

9.     Line 312, Please add (Figure 3) after “monolayer infections models”.

Author Response

Reviewer 2:

  1. With the exception of morphology assay, the authors did not use any controls of different species for other assays. It is not easy for readers to compare the results to known reported.

Answer:

Controls were used in each of the virulence assays, and the reference code for the ATCC strains has been included in the manuscript: C. albicans ATCC 10231 was used as a positive control for phospholipase activity and C. parapsilosis ATCC 22019 was used as a positive control for hemolytic activity. C. parapsilosis ATCC 22019 and C. albicans ATCC 10231 wereused as positive controls for protease activity. C. albicans ATCC 10231 was used as a positive control of biofilm production.

Controls were not used in chromogenic cultures because the behavior of the main Candida species in these media is well described.

  1. Figure 2, it would be reader friendly to have scale bars on the pictures such that the readers can appreciate the different size of cells between C. albicans and K. ohmeri. By doing so, it would be clear to know the magnification of the top and bottom panels of E1 and E2. 3. Figure 2, it would be nice to show photos of all three K. ohmeri isolates.

Answer:

The reviewer is right. It would be much more informative for the reader to have a way to compare the sizes of Kodamaeastructures to Candida, however, we do not have microscopes with which to determine the size of yeasts. This cannot be resolved and would constitute a limitation of the study.

  1. Figure 2, E1 of morphology of C. albicans seems like pseudohyphal form instead of true hyphal one.

Answer:

Indeed, Fig 2E1 shows the production of pseudohyphae by C. albicans.

  1. In addition to only three isolates, another limitation described by authors, the susceptibility testing could be done by broth microdilution method. In the discussion, the authors may add how to improve the detection of emerging species in their national surveillance system.  

Answer:

In the conclusions section the following is said: “On the other hand, our results demonstrate that the identification of emerging pathogens such as K. ohmeri is a challenge for clinical laboratories and that the implementation of tools based on molecular biology and/or mass spectrometry is urgently needed in the main hospital centers of the main cities of the country since some pathogens could be being misdiagnosed.” 

The authors consider that it is not advisable to expand the discussion on this topic because the main focus of the document could be lost or diluted, which is Kodamaea's first report in Honduras.

  1. There are several concerns as following.

Method: There may be a little confusion for readers at beginning if the paper related to K. ohmeri or C. auris.

Answer:

We appreciate the reviewer's observation. The materials and methods section explains the somewhat accidental rationale for discovering the three K. ohmeri isolates. The yeast surveillance program's goal is to identify and confirm Candida species, with a particular emphasis on detecting C. auris, as this emerging pathogen has not been reported in Honduras. Three isolates exhibited a restriction pattern (ITS/PCR RFLP) consistent with C. auris, but did not produce amplification with the especific primers for C. auris, prompting the need for sequencing. We have revised the M&M section as per your advice by shortening it and eliminating the description of Cauris's identification. The revised paragraph is as follows: “Identification of Candida auris. Isolates that produced bands of around 400 base pairs using primers ITS1 and ITS4, which is in line with the predicted pattern for the C. haemulonii complex and its related species C. auris [17], were chosen for a targeted PCR assay based on the GPI gene for the diagnosis of C. auris [18]. A band size of 137 bp indicated C. auris and no amplification product was expected for the rest of the Candida species. Isolates that presented PCR products of around 400 bp for the ITS marker, but failed to amplify the GPI gene, which is critical to confirming the presence of C. auris, were selected for sequencing”.

  1. Figure 2, it would be nice to show photos of all three ohmeriisolates. 

Answer:

Since the three Kodamaea isolates are morphologically identical, it would be redundant to include images of all three.

  1. If it is possible, please label the isolates with the contributing hospitals. That is, which one was from one hospital and the remaining two were from the other hospitals.

Answer:

The reviewer is correct; however, we do not have authorization from public officials to reveal the name of the hospitals from which the yeasts were isolated.

  1. Line 248, pleaseitalicize  auris.

Answer:

Done.

  1. Line 258, please describe as “identical” instead of “100% similarity”

Answer:

The sentence has been modified as requested.

  1. Line 312, Please add (Figure 3) after “monolayer infections models”.

Answer:

Done.

Round 2

Reviewer 2 Report

The revised one is good to be accepted.

The revised one is good to be accepted.